# Prompting Continual Person Search

## ABSTRACT

The development of person search techniques has been greatly promoted in recent years for its superior practicality and challenging goals. Despite their significant progress, existing person search models still lack the ability to continually learn from increasing real-world data and adaptively process input from different domains. To this end, this work introduces the continual person search task that sequentially learns on multiple domains and then performs person search on all seen domains. This requires balancing the stability and plasticity of the model to continually learn new knowledge without catastrophic forgetting. For this, we propose a **P**rompt-based **C**ontinual **P**erson **S**earch (PoPS) model in this paper. First, we design a compositional person search transformer to construct an effective pre-trained transformer without exhaustive pre-training from scratch on large-scale person search data. This serves as the fundamental for prompt-based continual learning. On top of that, we design a domain incremental prompt pool with a diverse attribute matching module. For each domain, we independently learn a set of prompts to encode the domain-oriented knowledge. Meanwhile, we jointly learn a group of diverse attribute projection and prototype embeddings to capture discriminative domain attributes. By matching an input image with the learned attributes across domains, the learned prompts can be properly selected for model inference. Extensive experiments are conducted to validate the proposed method for continual person search. The source code will be made available upon publication.

## CCS CONCEPTS

• **Computing methodologies → Object identification**.

## KEYWORDS

Visual Prompt, Continual Learning, Person Search

## 1 INTRODUCTION

Person search [56, 57, 66] aims to localize a target person in a gallery of uncropped scene images. It has attracted increasing research interest for its practicality and challenging goals. Existing works for person search have focused on boosting the performance under typical fully [1, 6, 30, 63] or weakly [13, 48, 58] supervised scenarios, and exploring domain adaptation [26] or generalization [36] methods. Despite their significant progress, these works learn only on a fixed and limited set of data while the real-world data is continually accumulating from different domains. To this end, we

Permission to make digital or hard copies of all or part of this work for personal or classroom use is granted without fee provided that copies are not made or distributed for profit or commercial advantage and that copies bear this notice and the full citation on the first page. Copyrights for components of this work owned by others than the author(s) must be honored. Abstracting with credit is permitted. To copy otherwise, or republish, to post on servers or to redistribute to lists, requires prior specific permission and/or a fee. Request permissions from permissions@acm.org.

*ACM MM, 2024, Melbourne, Australia*

© 2024 Copyright held by the owner/author(s). Publication rights licensed to ACM.
ACM ISBN 978-x-xxxx-xxxx-x/YY/MM
https://doi.org/10.1145/nnnnnnn.nnnnnnn

propose to explore the continual person search problem that learns from sequentially incoming domains and adaptively completes the person search task for any learned domain (see Figure).

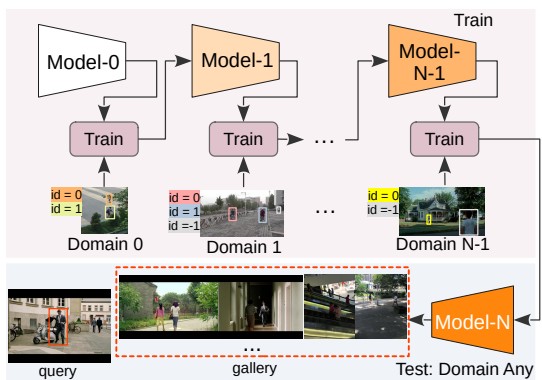

**Figure 1: Illustration of the continual person feature problem.**

A major challenge for enabling continual person search is to balance the stability and plasticity of the model to consistently adapt to new domains without catastrophic forgetting of seen domains. Recent works [12, 43, 53, 54, 54, 55] for continual image classification have drawn inspiration from visual prompt tuning [20] to employ a frozen pre-trained transformer [10, 47] to guarantee model stability and expandable visual prompts to encode domain-oriented knowledge for plasticity. In this way, the inference of the models relies on properly selecting learned prompts to classify an image from any seen domain. As the transformers are pre-trained to learn image-level object visual representations, it is natural to incorporate those models to tackle the classification tasks. However, the models are not compatible with person search as the task requires jointly localizing and extracting instance-level features of persons in the scene image. A straightforward solution for this is to collect large-scale scene images of persons and pre-train a re-designed person search transformer from scratch. Yet this can be expensive due to (1) collecting and annotating sufficient data, *e.g.* 14M images as in ImageNet-21K [8], and (2) performing large-scale pre-training which may require a dozen large-memory GPUs running for several days.

Besides, previous prompt-based continual learning methods mainly tackle the class incremental learning [12, 21, 43, 54, 55] and domain incremental learning [21, 53, 55] scenarios. Given that the former learns from sequential datasets with disjoint semantic space and the latter assumes all learning datasets share the same semantic space [46, 52], continual person search is more closely related to the domain incremental learning scenario. However, the learning domains [38] in those works are with clear boundaries (*e.g.* sketch image domain vs realistic image domain) which ease the adaptive selection of learned prompts during inference. In contrast, the domain gap between person search datasets can be ambiguous (*e.g.* both

CUHK-SYSU [56] and PRW [66] contain real-world images). This further raises a challenge to robustly capture the domain-specific attributes of an input image for properly selecting learned prompts to complete the person search task.

To tackle the aforementioned problems, we propose a **P**rompt-based **C**ontinual **P**erson **S**earch (PoPS) model in this work. Specifically, we design a compositional person search transformer that employs an existing hierarchical vision transformer, *e.g.* Swin [34], and expand the transformer with a Simple Feature Pyramid [28] to enable person localization. The vision transformer is pre-trained on the ImageNet-22K [8] dataset and is publicly available. We then only train the Simple Feature Pyramid on a moderate number of person detection data to form a pre-trained person search transformer. This makes better use of existing pre-trained transformers to reduce the consumption of energy and resources of large-scale pre-training from scratch. It also requires less data to optimize such a lightweight detection sub-network.

On top of the proposed person search transformer, we design a domain incremental prompt pool with diverse attribute matching to enable continual person search. The visual prompts are learned independently for each domain in the continual learning procedure similar to S-Prompts [53]. To capture domain-specific attributes for selecting learned prompts, we jointly learn a group of attribute projection and prototype embeddings. Similar to [43, 53–55], a pre-trained transformer is employed to extract the global feature of an input image as the query embedding. We then use the attribute projection embedding to uncover the domain attribute in the query embedding and learn to match the attribute with the correlated attribute prototype embedding. By enforcing the attribute projection and prototype embeddings to be diverse, this diverse attribute matching mechanism is capable of capturing discriminative domain attributes. Therefore, the correlated prompts can be selected by measuring the similarity between a query embedding and learned attributes across different domains.

To summarize, this paper makes the following contributions:

- We for the first time propose the continual person search problem. A Prompt-based Continual Person Search model is presented to consistently learn to adapt new person search tasks without catastrophic forgetting.
- By constructing a compositional person search transformer, we reduce the cost of pre-training for prompt-based continual person search from scratch. A domain incremental prompt pool with diverse attribute matching is proposed to adaptively reuse learned prompts by measuring the similarity between input images and learned attributes across domains.
- Extensive experiments are conducted to understand the effectiveness of the proposed modules for continual person search.

## 2 RELATED WORK

**Person Search.** The standard supervised learning of person search has been widely explored to achieve effective person search. Zheng et al. [66] first explores combining popular person detector and person re-identification (ReID) models for person search, resulting in a two-step mechanism that first detects and crops person images

and then retrieves a target across the cropped images. Following this mechanism, recent works obtained improved performance by enhanced person Re-ID features [5, 25] or designing target-conditioned person detectors [9, 50]. To improve the efficiency of the two-step paradigm, Xiao et al. [56] proposed an end-to-end method to perform person search by a unified model. Following works [4, 6, 15, 45] further explored to effectively balance the multiple training objectives as one-step person search is a multi-task learning problem. Other methods [33, 35, 61] employed the context prior knowledge to match different persons across images. It is also practical to improve model efficiency by introducing lightweight detector architectures [60, 65], or boost the model performance by designing a stronger detection sub-network [30]. Inspired by recent advances in vision transformers [10, 47, 67], recent works [1, 63] obtained more discriminative person features with well-designed person search transformers.

As the manual annotation of person search data is extremely expensive, weakly supervised person search is proposed to train a person search model with only person bounding box annotations. To enable weakly supervised person search in a one-step manner, recent works [14, 49, 59] proposed to generate pseudo identity labels by clustering and then learn person feature representations in a supervised way during each training epoch. Alternatively, this task can be completed by a two-step model composed of a fully supervised person detector and an unsupervised person ReID model [19]. To enable unsupervised domain adaptative person search that pre-trains on a labeled source domain and then adapts to an unlabeled target domain, Li et al. [27] designed both image-level and instance-level domain alignment modules to narrow the domain gap. Oh et al. [37] further proposed to learn from unreal data to facilitate domain generalizable person search.

**Prompt-based Continual Learning.** To enable continual learning without a rehearsal buffer [3, 16], Wang et al. [55] proposed the first prompt-based continual learning method that designs a prompt pool with paired keys and prompts. The prompts help a pre-trained transformer to adapt to new tasks by visual prompt tuning [20], and the keys are learned to adaptively pick prompts for an input image. Based on this, DualPrompt [54] jointly learned task-specific expert prompts and task-shared general prompts for prompt tuning. CODA-P [43] introduced a set of prompt components and implicitly learned the attention weight for fusing the prompt components, allowing adaptive weighted prompt summation instead of selection. LGCL [23] further introduced language guidance to learn a unified semantic embedding space for continual classification. DAP [22] instead learned a prompt generation module to construct a pool-free approach. Other works [44, 51] analyzed the effect of pre-trained transformers and boosted the performance when using self-supervised pre-trained [2, 7] or moderate pre-trained transformers. Different from these works that mainly tackle class incremental learning [46, 48], S-Prompts [53] designed a simple yet effective method for domain incremental learning [46, 52]. After independently performing visual prompt tuning [20] on a seen domain, S-Prompts used K-Means to memorize data centroids of the domain and infer the domain ID by assigning an input image with its nearest centroid. This inspires us to design a domain incremental prompt pool as the continual person search is more closely to the domain incremental learning task.

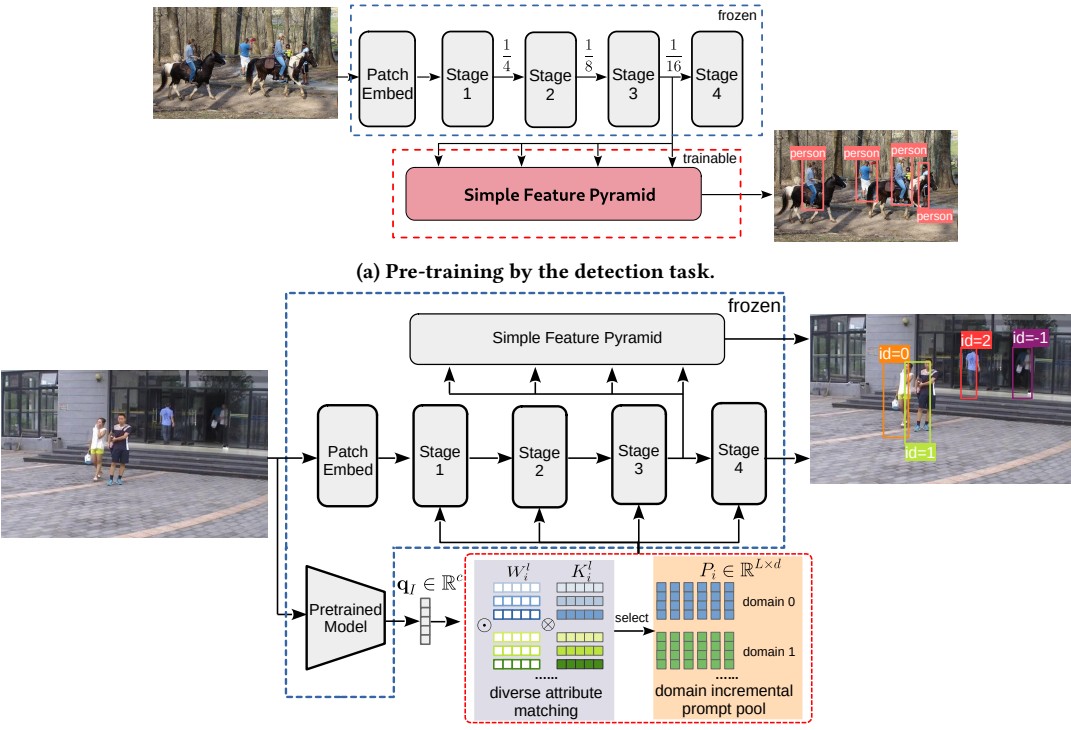

(a) Pre-training by the detection task.

(b) Continual learning for person search tasks.

Figure 2: The overall architecture of the proposed Prompt-based Continual Person Search Model. We construct a person search network from a pre-trained Swin Transformer [34] with a Simple Feature Pyramid [28]. This allows (a) preparing a pre-trained person search transformer by an additional person detection task instead of large-scale pre-training from scratch. On top of that, we conduct (b) prompt-based continual learning of person search tasks by designing a prompt pool with diverse attribute matching. This exploits the prior knowledge encoded by the pre-trained model and balances the stability and plasticity for continual learning.

## 3 METHOD

### 3.1 Overview

The overall architecture of the proposed Prompt-based Continual Person Search model is depicted in Figure 2. We at first design a compositional person search transformer by expanding a hierarchical vision transformer [34] pre-trained on the widely-used ImageNet [8] data with a Simple Feature Pyramid [28] (see Figure 2a). As the vision transformer is naturally capable of extracting person visual features, this enables the overall model to localize appeared persons in scene images and forms an effective person search network. As the continual person search problem is more closely related to the domain incremental learning task [46, 52], we propose to solve the problem by learning domain incremental visual prompts. Specifically, on top of the proposed person search network, we design a domain incremental prompt pool [43, 53–55] that independently learns visual prompts correlated with seen domains. By introducing diverse learnable attribute projection and prototype embeddings, the prompt pool learns to capture diverse domain attributes to select proper prompts at the test time (see Figure 2b).

On top of the designed modules, the overall continual learning procedure can be completed in two stages: (1) pre-training on a person detection task (Figure 2a) and (2) continual learning on incoming person search tasks (Figure 2b). In the first stage, we fix the pre-trained vision transformer and only optimize the Simple Feature Pyramid on a set of person detection data to learn prior knowledge for person detection. We thereby obtain a pre-trained person search transformer for the subsequent continual learning stage without exhaustive pre-training. This makes better use of existing pre-trained models and reduces the resource consumption of pre-training from scratch. In the second stage, we fix the overall pre-trained model and learn a small set of visual prompts for each seen person search domain by visual prompt tuning [20]. By incrementally learning domain-oriented visual prompts and properly selecting prompts at test time, the proposed method is capable of balancing the stability and plasticity for continual learning.

In the following content, we first describe how the compositional person search transformer process input scene images to complete the task in subsection 3.2, and then introduce the person detection pre-training and person search continual learning stages in subsections 3.3 and 3.4, respectively.

## 3.2 Compositional Person Search Transformer

Largely pre-trained vision transformers [10, 47] are vital for prompt-based continual learning [43, 53–55]. Yet the models are not directly applicable to the person search task for the lack of person localization modules. It is also time- and resource-consuming to reformulate and pre-train a person search transformer from scratch. To this end, we employ a typical hierarchical vision transformer, *i.e.* Swin [34], pre-trained on the ImageNet [8] data and expand the transformer with a Simple Feature Pyramid [28] to enable person localization.

As in Figure 2a, an input image $\mathbf{I} \in \mathbb{R}^{H_0 \times W_0 \times 3}$ is partitioned into multiple equally-sized patches and each patch is projected into a high-dimensional vector by the 'Patch Embed' layer, resulting in an intermediate image feature map $\mathbf{F}_{img} \in \mathbb{R}^{H \times W \times C}$. Afterward, Swin introduces 4 stages where each stage performs downsampling and consecutive window-based multi-head self-attention [47] on their inputs to produce deep visual representations. On top of that, we introduce the Simple Feature Pyramid [28] to enable the overall model for person localization. Following [28], we build the Simple Feature Pyramid on the 16 times downsampled feature map $\mathbf{F}'_{img} \in \mathbb{R}^{H' \times W' \times C'}$, *i.e.* the output of 'Stage3'. By applying convolutions of strides $\{2, 1, \frac{1}{2}, \frac{1}{4}\}$ on $\mathbf{F}'_{img}$ in parallel, where the fractional strides indicate deconvolutions, we obtain image feature maps of scales $\{\frac{1}{32}, \frac{1}{16}, \frac{1}{8}, \frac{1}{4}\}$. Based on the multi-scale features maps, we follow Mask R-CNN [17] to predict person bounding boxes and detection confidences.

Different from the image classification [11, 34] or person re-identification tasks [62, 64] that directly extract object features from object-centered images, the person search task requires to extract features of instances in the scene images. To this end, we dissect Swin into two parts, *i.e.* 'Patch Embed' to 'Stage3' blocks that process the integral image feature maps, and the 'Stage4' block that refines instance-wise person feature maps. For illustration purposes, we exemplify the person feature extraction procedure in Figure 3. According to the person bounding boxes predicted by the detection sub-network, we use ROIAlign [17] to extract interpolated feature maps within the boxes as person feature maps. We then employ the 'Stage4' block on the extracted feature maps to obtain refined person feature maps $\mathbf{f}_{roi} \in \mathbb{R}^{h \times w \times d}$. Afterward, we follow the practice in Swin to conduct global average pooling on $\mathbf{f}_{roi}$ and use the pre-trained Layer Normalization layer to produce final 1D person features $\mathbf{v} \in \mathbb{R}^d$.

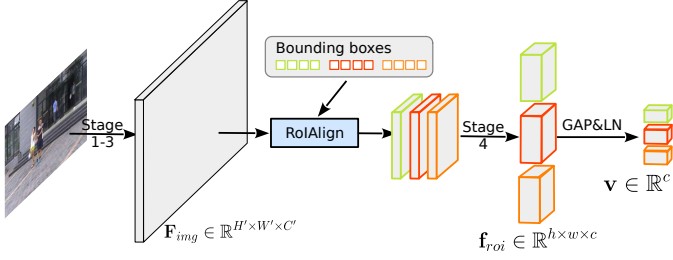

**Figure 3: Illustration of person feature extraction.**

## 3.3 Pre-training by Person Detection

Although the compositional person search transformer makes a pre-trained transformer applicable for person search, the newly added modules are randomly initialized and thus contain no prior knowledge for subsequent continual learning. To tackle this problem, we propose to pre-train only the detection sub-network to construct a pre-trained person search transformer. As the pre-trained Swin [34] is frozen in this stage, the pre-trained visual feature space is left unchanged for person feature extraction and the detection sub-network only needs to predict person locations from the pre-trained image features. It is also worth noting that the number of the learnable parameters in this mechanism is relatively small, thus the pre-training can be completed with less data.

Specifically, we combine the training set of CrowdHuman [41] and images containing humans in the training set of MSCOCO [32] to form a person detection dataset similar to Shuai et al. [42]. In total, this collection presents 79,115 scene images that contain nearly 0.6M person instances for training the model. Compared with the pre-training (using ImageNet-21K [8] that contains 14M images) of the widely used vision transformers in prompt-based continual learning methods [43, 53–55], the proposed pre-training by person detection is more data-efficient.

Based on the collected data, we pre-train the model by conducting person detection to form a pre-trained person search transformer. As is mentioned in Subsection 3.2, we employ the 'Patch Embed' to 'Stage3' blocks of the pre-trained Swin [34] to extract image feature maps and send the feature maps to the detection sub-network to detect appeared persons (see Figure 3.3). The overall training objective is formulated as

$$\mathcal{L}_{det} = \mathcal{L}_{reg} + \mathcal{L}_{cls} + \mathcal{L}_{reg}^{rpn} + \mathcal{L}_{cls}^{rpn} \tag{1}$$

where $\mathcal{L}_{reg}$ and $\mathcal{L}_{cls}$ are the bounding box regression and classification losses of the detection head [17], $\mathcal{L}_{reg}^{rpn}$ and $\mathcal{L}_{cls}^{rpn}$ are those of the Region Proposal Network [40].

We also note that the data used for this pre-training stage shares similar spirits with that in [42]. However, Shuai et al. [42] employs the data to pre-train a convolutional person search network by a self-supervised person similarity learning framework. The pre-trained model can be fine-tuned for a specific downstream task to achieve improved person search performance. Different from Shuai et al. [42], we focus on efficiently pre-training a person search transformer by a person detection task with the collected data. The pre-trained model is used to enable prompt-based continual learning without fine-tuning.

## 3.4 Continual Learning for Person Search

A key target in continual learning is to balance the stability and plasticity of the model for absorbing knowledge of new tasks without forgetting that of learned tasks. For this, recent advanced continual learning methods [21, 43, 53–55] have explored a prompt-based mechanism that exploits a largely pre-trained vision transformer with an incrementally learned prompt [20] pool. In this way, the pre-trained transformer is fixed and a set of learnable prompts are injected into the input sequence to the self-attention layer [10, 47] to adapt to a new task. Thus the continual learning problem can be solved by continually learning new visual prompts for new tasks

during training and adaptively reusing proper prompts during inference. Inspired by this mechanism, we design a domain incremental prompt pool with diverse attribute matching to enable continual person search.

**Domain incremental prompt pool.** On top of the previously pre-trained compositional person search transformer, we independently learn domain-oriented visual prompts during continual learning similar to S-Prompts [53]. Specifically for a domain $i$, we set a sequence of randomly initialized learnable prompts $P_i^l \in \mathbb{R}^{L \times d}$, where $L$ is the number of prompts, for the $l$-th self-attention layer as in Figure 2b. Different from previous works [43, 53–55] that employ ViT [10] with global self-attention layers, Swin [34] partitions the input feature map into local windows and performs window-based self-attention [34] to reduce the computation complexity. Thus for the $l$-th Swin transformer layer, we duplicate the correlated prompts $P_i^l$ for each partitioned window and conduct self-attention as

$$\hat{z}^l = \text{MHSA}(\text{CAT}(P_i^l, z^{l-1}))[L:] \tag{2}$$

where MHSA refers to multi-head self-attention and CAT is the concatenation operation. $z^{l-1}$ is the flattened image feature map within a local window. As MHSA keeps the length of the input sequence, we use $[L:]$ to preserve only the output corresponding to the input image feature tokens, leaving the spatial shape of the overall image feature maps unchanged for subsequent blocks. By applying the learnable prompts, we independently train the model with the detection loss in Equation 1 and the widely used Online-Instance-Matching (OIM) loss [56] $\mathcal{L}_{oim}$ for each incoming person search domain.

**Diverse attribute matching.** As the domain identity of a test task is unavailable during inference, the learned visual prompts should be adaptively selected for inference. To tackle this problem, we design a diverse attribute matching module to measure the similarity between an input image and seen domains. Following previous prompt-based continual learning methods [43, 53–55], we duplicate the pre-trained Swin [34] to extract the global visual feature as $q_i = F(I_i), q_i \in \mathbb{R}^c$ of an input image $I_i$ belonging to domain $i$. As the pre-trained model is agnostic to the incoming person search tasks, the visual feature $q_i$ implicitly encodes unbiased attributes of domain $i$. To capture discriminative domain features, i.e. domain attributes, we bind a group of $N$ learnable embeddings $K_i^l = \left\{ k_i^j \in \mathbb{R}^c | j = 1, 2, \dots N \right\}$ with the visual prompts $P_i^l$ as attribute prototypes of domain $i$. By maximizing the similarity between $q_i$ and $k_i^j$ similar to L2P [55], the attribute embeddings are optimized to match the attributes of domain $i$ during training and the learned prompts can be selected by matching the input image with learned domain attributes during inference. However, this may cause redundancy between different attribute embeddings as the multiple learnable prototypes can easily overfit the same prominent domain attribute. As the domain scenarios can be highly similar between different person search datasets, it also requires learning more diverse domain attributes to mostly uncover the differences between different domains.

For this, we further attach a group of learnable attribute projection embeddings $W_i^l = \left\{ w_i^j \in \mathbb{R}^c | j = 1, 2, \dots N \right\}$ to $K_i^l$. $W_i^l$ can be regarded as learned channel attention [18] to emphasize a certain

attribute of domain $i$ in an input image. For training on domain $i$, given an input scene image $I_i$, we calculate the similarity between the image global feature $q_i = F(I_i)$ and the attribute prototype $k_i^j$ as

$$a_i^j = q_i \odot w_i^j \otimes k_i^j \tag{3}$$

where $\odot$ denotes channel-wise multiplication and $\otimes$ stands for calculating cosine similarity. The domain attribute learning loss is thus formulated as

$$\mathcal{L}_{attr} = \sum_{j=1}^{N} \left( 1 - a_i^j \right) \tag{4}$$

Besides, both $\left\{ w_i^j | j = 1, 2, \dots N \right\}$ and $\left\{ k_i^j | j = 1, 2, \dots N \right\}$ are enforced to be diverse by a diversity loss

$$\mathcal{L}_{div} = \sum_{m=1}^{N} \sum_{n=1, n \neq m}^{N} |w_i^m \otimes w_i^n|^2 + \sum_{m=1}^{N} \sum_{n=1, n \neq m}^{N} |k_i^m \otimes k_i^n|^2. \tag{5}$$

The overall training objective on domain $i$ is thus given by

$$\mathcal{L}_i = \mathcal{L}_{det} + \mathcal{L}_{oim} + \lambda_1 \mathcal{L}_{attr} + \lambda_2 \mathcal{L}_{div} \tag{6}$$

where $\lambda_1$ and $\lambda_2$ are predefined loss weights.

During inference, we calculate the similarity between an input image $I$ and a learned domain as

$$s_{I \to i} = \frac{1}{N} \sum_{j=1}^{N} q_I \odot w_i^j \otimes k_i^j \tag{7}$$

where $q_I = F(I)$. Thus the domain index $d$ of the selected prompts $P_d^l$ is given by

$$d = \arg\max_i (\{ s_{I \to i} | i = 1, 2, \dots, D \}) \tag{8}$$

where $D$ is the number of learned domains.

We also note that the learning of diverse domain attributes shares a similar technique with CODA-P [43]. Yet CODA-P is designed for class incremental learning while our work deals with a domain incremental learning problem. CODA-P trains without maximizing the matching scores but implicitly learns attention weights to fuse all prompt components through visual prompt tuning [20]. Yet we explicitly optimize the matching scores to guarantee adaptively selecting of learned prompts. The attention mechanism in CODA-P [43] can also be included in the proposed method to further boost the continual learning performance.

## 4 EXPERIMENTS

### 4.1 Datasets and Evaluation Protocol

**CUHK-SYSU** [56] collected 18,184 images from both movies and real-world street snapshots, presenting 96,143 bounding boxes of pedestrians and 8,432 labeled identities in total. The training set contains 11,206 frames with 5532 identities. The testing set selects 2900 query persons and defines different evaluation protocols with varied gallery sizes.

**PRW** [66] collected scene images from 6 cameras deployed at a campus. In total, it presents 11,816 frames containing 43,110 pedestrian bounding boxes with 932 recognizable identities. The training

subset contains 5,134 frames with 432 identities and the test set contains 6,112 frames. Different from CUHK-SYSU [56], the evaluation protocol of PRW takes the full test set as the gallery.

**MovieNet-PS** [39] selected 160K frames of 3,087 identities from 385 movies. The training set keeps persons of 2,087 identities and a test set is with the left 1,000 identities. For training, it presents 3 different settings that preserve at most 10, 30, and 70 instances per identity, resulting in 20K, 54K, and 100K training images in total. For evaluation, the gallery set is constructed in a way similar to [56] with varying sizes.

**Evaluation protocol.** For experiments of continual person search, we sequentially train the person search model with the three datasets to simulate continual learning on incoming domains and then test the model performance on each learned domain without knowing the domain ID. Similar to previous person search research [1, 6, 30, 56, 63], we use the **mAP** and **top-1** metrics to evaluate the person search accuracy and present the results in the format of **mAP** / **top-1** in the following experiments. The **AP** and **Recall** scores are used to evaluate the model for person detection. Moreover, the gallery size is closely related to the degree of retrieval difficulty in person search. To examine the overall model performance, we calculate the weighted average of a person search performance metric as

$$M_{avg} = \frac{G_c}{G} M_c + \frac{G_p}{G} M_p + \frac{G_m}{G} M_m, G = G_c + G_p + G_m \quad (9)$$

where $M_c$, $M_p$, and $M_m$ are the performance metrics on CUHK-SYSU, PRW, and MovieNet-PS, respectively. $G_c$, $G_p$ and $G_m$ are the default gallery size of the respective datasets. We also measure the forgetting of the metrics on learned domains in the same way. Formally, the forgetting measurement is given by the performance decay on domains $i (i < D)$ after learning on domain $D$ compared with that when complete learning on domain $i$. By default, the continual learning sequence is set to CUHK-SYSU [56]→PRW [66]→MovieNet-PS [39].

## 4.2 Implementation Details

For the compositional person search transformer, we use Swin-S [34] pre-trained on ImageNet-22K [8] as the pre-trained vision transformer. The output size of RoIAlign [17] is set to $14 \times 14$ to match the size of image features during pre-training. For the pre-training on person detection data, the model is trained for 36 epochs with a batch size of 16. The input image is augmented with random horizontal flip and scaling where the shorter side varies from 480 to 800 during training. We use the AdamW optimizer with an initial learning rate of 0.0001. The learning rate is linearly warmed up during the first 1,000 iterations and decreased by 10 at the $20^{th}$ epoch. During the continual learning stage, we employ the Adam optimizer with an initial learning rate of 0.0003. For CUHK-SYSU [56] and PRW [66], we keep the image augmentation with batch size 8 and resize the images to $1333 \times 800$ for evaluation. For MovieNet-PS [39], we instead randomly resize the shorter image side from 160 to 240. The test image size is set to $720 \times 240$. Given the unbalanced scales of the datasets, we train the proposed PoPS model for approximately equal steps, 22k iterations, on different domains (*i.e.* 16 epochs on CUHK-SYSU [56] and MoviNet-PS [39], 32 epochs on PRW [66]). The loss weights $\lambda_1$ and $\lambda_2$ in Equation

6 are both set to 0.1. The test gallery sizes of CUHK-SYSU and MovieNet-PS are 100 and 2000, respectively.

## 4.3 Continual Person Search

To validate the effectiveness of the proposed PoPS method, we conduct continual person search evaluation on *three* types of compared methods: (1) sequentially fine-tuned models including 'Prompt + FT-seq' that combines the proposed compositional person search transformer with fixed length prompts, and a representative previous person search model SeqNet [30] with the Swin [34] backbone used in our method; (2) representative continual learning methods applying to the proposed compositional person search transformer; (3) upper-bound model that performs prompt tuning [20] with the compositional person search transformer on the union of all domains. We also test to incorporate the attention mechanism [43] into PoPS. We directly replace each prompt with a prompt component and bind extra attention vectors and keys [43] while the weighted summation of prompt components is restricted in the same domain.

The experimental performances of the aforementioned methods are presented in Table 1. It can be observed that our proposed method obtains the best overall accuracy for continual person search. The anti-forgetting performance is also shown to be superior. In addition, incorporating the attention mechanism in CODA-P [43] further improves the accuracy by introducing more learnable parameters. Compared with previous continual learning methods, sequential prompt tuning of the compositional person search transformer performs only slightly inferior on forgetting learned knowledge, while the sequentially fine-tuned SeqNet [30] is marginally behind. We believe the frozen transformer mainly improves the model stability. We also observe that the domain incremental learning methods S-Prompt [53] and our proposed PoPS perform significantly better on anti-forgetting of the PRW [66] domain, demonstrating the superior of domain incremental learning techniques for continual person search.

## 4.4 Analytical Studies

In this subsection, we conduct ablation experiments to understand the impact of the designed modules in PoPS for continual person search. For the convenience of notation, we use **MVN** as the short for **MovieNet-PS**.

**The effectiveness of detection pre-training.** To construct an effectively pre-trained person search transformer without exhaustive pre-training from scratch, we design a compositional person search transformer by expanding a pre-trained Swin with a Simple Feature Pyramid. The added sub-network is then pre-trained by a person detection task. To examine the effectiveness of detection pre-training, we test the pre-trained model on the three person search datasets for person detection and collect the results in Table 2. We also test a standard detector Faster R-CNN [40] independently trained on the datasets for comparisons. Although the pre-trained PoPS is evaluated in a cross-domain manner, it can be observed the person detection result is still acceptable compared with the fully supervised Faster R-CNN [40]. This suggests that the detection pre-training is effective for preparing a person search transformer for prompt-based continual learning.

**Table 1: Continual person search performance comparisons between our proposed PoPS and existing methods. We collect both the person retrieval accuracy and forgetting metrics to make a comprehensive understanding of the effectiveness of PoPS. All results are given as mAP / top-1.**

| Method | Accuracy (↑) | | | | Forgetting (↓) | | |
|---|---|---|---|---|---|---|---|
| | CUHK-SYSU | PRW | MovieNet-PS | Average | CUHK-SYSU | PRW | Average |
| Prompt + FT-seq | 83.2 / 85.3 | 20.4 / 76.8 | 29.6 / 81.0 | 23.4 / 77.9 | 9.9 / 8.8 | 25.7 / 7.6 | 25.4 / 7.6 |
| SeqNet [30] (Swin [34])-seq | 75.6 / 77.3 | 19.9 / 75.6 | 34.8 / 81.3 | 24.2 / 77.0 | 17.8 / 16.8 | 25.9 / 9.1 | 25.8 / 9.2 |
| L2P [55] | 85.0 / 87.1 | 21.7 / 77.0 | 30.0 / 81.3 | 24.5 / 78.2 | 7.8 / 6.7 | 24.3 / 7.1 | 24.0 / 7.1 |
| DualPrompt [54] | 83.5 / 84.9 | 22.1 / 77.5 | 30.1 / 81.0 | 24.8 / 78.4 | 9.3 / 8.7 | 23.6 / 7.8 | 23.4 / 7.8 |
| CODA-P [43] | 84.9 / 85.7 | 26.2 / 76.6 | 32.3 / 82.2 | 28.4 / 78.1 | 9.7 / 9.0 | 26.8 / 9.3 | 26.5 / 9.3 |
| S-Prompt [43] | 85.6 / 87.4 | 39.7 / 81.5 | 24.8 / 76.5 | 36.6 / 80.4 | 7.2 / 6.3 | 6.3 / 2.6 | 6.3 / 2.7 |
| **PoPS** | 86.3 / 87.3 | 42.8 / 83.3 | 25.4 / 77.3 | 39.1 / 81.9 | **5.8 / 6.0** | **0.1 / 0.2** | **0.2 / 0.3** |
| **PoPS** + Attention [43] | **87.5 / 88.6** | **49.6 / 85.8** | 26.8 / 79.5 | **44.5 / 84.3** | 6.8 / 6.4 | 0.2 / **0.2** | 0.3 / **0.3** |
| Prompt + upper-bound | 91.6 / 92.8 | 46.3 / 84.1 | 40.1 / 86.4 | 45.4 / 84.8 | - | - | - - |

**Table 2: Person detection performance comparisons between the pre-trained compositional person search transformer and a standard Faster R-CNN [40] detector.**

| Method | CUHK-SYSU | | PRW | | MVN | |
|---|---|---|---|---|---|---|
| | AP | Recall | AP | Recall | AP | Recall |
| PoPS | 81.8 | 88.0 | 90.0 | 95.4 | 73.7 | 83.3 |
| Faster R-CNN [40] | 87.0 | 92.9 | 93.0 | 96.1 | 89.4 | 96.9 |

**The effect of Simple Feature Pyramid.** In the proposed compositional person search transformer, we introduce the Simple Feature Pyramid (SimFPN) [28] to enable person detection. Yet it is also compatible to employ the standard Feature Pyramid Network (FPN) [31]. For this, we compare the continual person search performance with different feature pyramids as in Table 3. The results suggest that the Simple Feature Pyramid enables superior continual person search, especially on the anti-forgetting of previously learned domains. We hypothesize the main reason is that SimFPN [28] is derived from a deep intermediate feature and better encodes the prompts injected in previous layers.

**Table 3: The continual person search performance of PoPS when employing different feature pyramids.**

| Det. | CUHK-SYSU | PRW | MVN | Average |
|---|---|---|---|---|
| FPN [31] | 76.7 / 78.4 | 41.6 / 82.5 | 25.3 / 75.8 | 38.1 / 80.8 |
| SimFPN [28] | 86.3 / 87.3 | 42.8 / 83.3 | 25.4 / 77.3 | 39.1 / 81.9 |

**The effect of attribute projection.** To encourage diverse domain attribute learning for adaptive prompt selection, we introduce the attribute projection embeddings $W_i^l$ to reveal the domain attribute information in an input image before matching it with the learned attribute prototypes. To validate the effect of the attribute projection embeddings, we test PoPS with and without them as in Table 4. Without the attribute projection embeddings, we directly maximize the similarity between the image feature $\mathbf{q}_i$ and its

correlated diverse attribute prototypes $K_i^l$ during training. It can be observed that introducing the attribute projection consistently improves model performance on early domains, suggesting that this mechanism enables more robust prompt selection.

**Table 4: Continual person search performance comparisons between PoPS with and without the attribute projection.**

| Proj. | CUHK-SYSU | PRW | MVN | Average |
|---|---|---|---|---|
| ✗ | 81.3 / 82.3 | 40.9 / 82.0 | 25.4 / 77.1 | 38.4 / 81.2 |
| ✓ | 86.3 / 87.3 | 42.8 / 83.3 | 25.4 / 77.3 | 39.1 / 81.9 |

**Comparison between deep prompt and shallow prompt.** As the vision transformers [10, 34] are formulated in a multi-layer architecture, it is also important to explore which layers to insert the prompts. For this, VPT [20] test both shallow (prompts for only the first layer) and deep (prompts for every layer) prompts with ViT [10]. Different from ViT [10], Swin [34] is composed of 4 stages where each of them contains several layers. We therefore test shallow and deep prompts upon stages for PoPS as in Table 5. The results suggest that the model requires deep stage-wise prompts to achieve the best performance. Moreover, adding prompts to the last stage gives a significant improvement in the performance. This is mainly due to that 'Stage4' processes the features in a different way, *i.e.* instance-wise, thus requiring extra prompts to adapt well to the person search task.

**Table 5: Continual person search performance of PoPS when prompting different Swin [34] stages.**

| Stages | CUHK-SYSU | PRW | MVN | Average |
|---|---|---|---|---|
| (1,) | 67.6 / 69.4 | 33.6 / 65.3 | 19.9 / 60.6 | 30.7 / 64.2 |
| (1,2) | 69.1 / 70.6 | 34.3 / 66.7 | 20.3 / 61.9 | 31.3 / 65.6 |
| (1,2,3) | 72.6 / 73.8 | 36.1 / 70.0 | 21.3 / 64.9 | 32.9 / 68.6 |
| (1,2,3,4) | 86.3 / 87.3 | 42.8 / 83.3 | 25.4 / 77.3 | 39.1 / 81.9 |

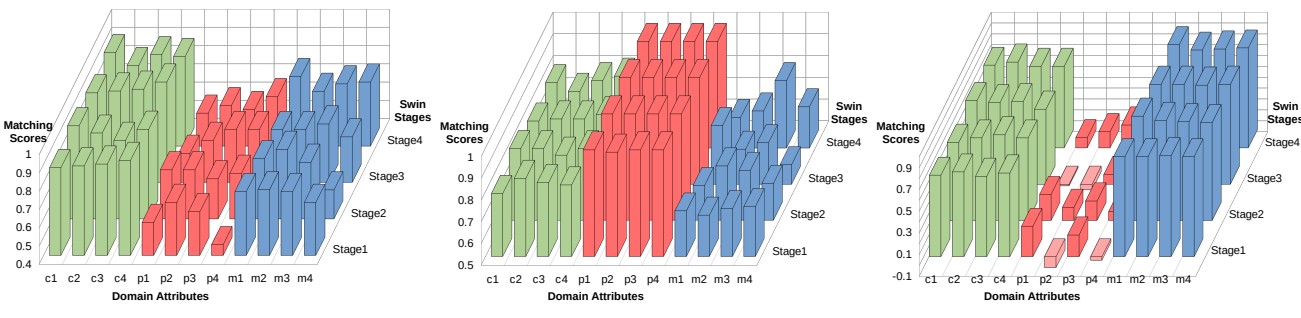

(a) Attribute matching on CUHK-SYSU [56].    (b) Attribute matching on PRW [66].    (c) Attribute matching on MVN [39].

Figure 4: Exampler visualization of prompt selection on CUHK-SYSU [56], PRW [66] and MVN [39]. We denote by c$i$, p$i$ and m$i$ the learned $i$-th domain attribute from the three datasets, respectively.

**The effect of the number of prompts.** With the person search transformer frozen during training, the number of learnable prompts for each transformer layer is also important. To validate the effect of different numbers of learnable prompts, we conduct the experiments in Table 6. By default, we set the number of prompts $L$ to 16. It can be observed that decreasing the prompts leads to a significant drop in model performance while adding more prompts only slightly improves the results. As the window attention performs self-attention [47] on relatively short image feature sequences (e.g. $7 \times 7$), we assume that the inserted prompts should be well balanced with the image feature. Thus the optimal number of prompts should be moderate.

Table 6: Continual person search performance of PoPS with different numbers of learnable prompts.

| Stages | CUHK-SYSU | PRW | MVN | Average |
|---|---|---|---|---|
| 4 | 81.3 / 82.3 | 39.9 / 82.1 | 23.6 / 75.3 | 36.4 / 80.4 |
| 8 | 84.7 / 86.0 | 40.7 / 82.1 | 24.1 / 75.9 | 37.2 / 80.6 |
| 16 | 86.3 / 87.3 | 42.8 / 83.3 | 25.4 / 77.3 | 39.1 / 81.9 |
| 32 | 86.5 / 87.8 | 43.3 / 83.6 | 26.0 / 77.5 | 39.6 / 82.2 |

**The impact of the number of learnable domain attributes.** To properly select learned prompts for continual person search, we learn to capture $N$ diverse attributes for recognizing each domain. To validate the impact of $N$, we test the continual person search performance of PoPS with varied $N$ as in Table 7. It can be observed that learning 4 diverse domain attributes results in better overall performance. When setting a small $N$, the model may fail to capture sufficient distinct domain features. And the attribute embeddings may overfit on training samples when setting a large $N$ and impedes the person search performance We thus set $N = 4$ for each domain by default.

**Visualization of prompt selection.** To qualitatively understand the effectiveness of adaptive prompt selection with diverse attribute matching, we visualize the matching scores between input images and learned attributes across different domains. The scores are averaged on 10 randomly selected test images from the learned domains as in Figure 4. We also average the scores in different layers within the same stage to obtain stage-level matching scores. It can

Table 7: Evaluation of continual person search for PoPS with different numbers of learnable domain attributes.

| N | CUHK-SYSU | PRW | MVN | Average |
|---|---|---|---|---|
| 2 | 85.7 / 86.9 | 42.5 / 82.4 | 22.1 / 74.1 | 38.1 / 80.4 |
| 4 | 86.3 / 87.3 | 42.8 / 83.3 | 25.4 / 77.3 | 39.1 / 81.9 |
| 8 | 84.9 / 86.3 | 40.7 / 81.9 | 22.4 / 75.0 | 36.8 / 80.3 |

be observed that the matching scores between the test image and the truly corresponding domain attributes are relatively high compared with the distractors, suggesting the effectiveness of diverse attribute matching. As the learning domains can be highly similar in continual person search, we also observe that there always exists a highly similar distracting domain when testing the model.

## 5 CONCLUSION

This work introduces a new challenging yet practical continual person search task that learns from sequentially incoming domains and adaptively completes the person search task for any learned domain. To better balance the stability and plasticity of the model to consistently adapt to new domains without catastrophic forgetting of seen domains, we design a Prompt-based Continual Person Search model (PoPS). To reduce the cost of pre-training a person search transformer from scratch, we first propose a compositional person search transformer that expands a pre-trained hierarchical vision transformer with a Simple Feature Pyramid to enable person localization. We then pre-train the Simple Feature Pyramid with only a moderate number of person detection data to construct a fully pre-trained person search transformer. On top of that, we design a domain incremental prompt pool with a diverse attribute matching module. During training, the prompts are independently learned to encode the domain-oriented knowledge, and a group of paired attribute projection and prototype embeddings are forced to diversely capture distinct domain attributes. This facilitates the adaptive selection of learned prompts by matching an input image with the learned attributes across domains for model inference. For future works, we shall explore designing a more efficient continual learning framework and collecting more realistic domains to better tackle the continual person search problem.

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
