# OpenReview forum: "Prompting Continual Person Search"
_acmmm.org/ACMMM/2024/Conference — MM2024 Poster_

### Official Review · Reviewer_uZNe · 2024-05-24

**Rating:** 3
**Confidence:** 3

**Summary:**

This work introduces the continual person search task that sequentially learns on multiple domains and then performs person search on all seen domains. For this, they propose a Prompt-based Continual Person Search (PoPS) model in this paper. They design a compositional person search transformer to construct an effective pre-trained transformer without exhaustive pre-training from scratch on large-scale person search data. On top of that, they design a domain incremental prompt pool with a diverse attribute matching module.

**Strengths:**

1. This work is the first to propose person search in a continual learning scenario, which is somewhat pioneering.
2. This paper leverages existing methods in person search and the paradigm of prompt learning in continual learning to address this problem. The overall framework is relatively concise and reasonable.
3.  The writing is clear and easy to understand, and the experiments are relatively comprehensive.

**Limitations:**

1.  Since this paper is the first to propose this task, theoretically, it should formalize the definition of this task. However, this part is missing in the main text or supplementary materials.
2. I believe the key issue with this paper is whether this topic has genuine significance. In fact, there are already many methods for person re-identification in continual learning scenarios [1][2][3]. Therefore, the continual person search proposed in this paper may be decomposed into a person detection task and a continual person re-identification task, each of which can be addressed using existing methods. As mentioned in Table 2 of this paper, the performance of PoPS is not even as good as Faster R-CNN in person detection. Therefore, if we were to introduce a detector stronger than Faster R-CNN and combine it with an existing Lifelong Person Re-identification method, could we achieve higher performance than PoPS? However, the authors do not seem to cite any of the aforementioned Lifelong Person Re-identification methods in the paper, nor do they discuss the fundamental differences between this task and the combination of person detection and Lifelong Person Re-identification.

I would consider raising the score if the authors can persuade me with theory or experimental results during the rebuttal phase.

[1] Prompt Based Lifelong Person Re-identification, PRCV 2023
[2] Patch-based Knowledge Distillation for Lifelong Person Re-Identification, ACM MM 2022
[3] Lifelong person reidentification via adaptive knowledge accumulation, CVPR 2021

**Suitability:**

2

---

### Official Review · Reviewer_Swey · 2024-05-25

**Rating:** 3
**Confidence:** 2

**Summary:**

The paper develops the continual person search task, which aims to localize a target person across different domains. It designs a frozen swin vision transformer with a simple feature pyramid for person localization, and a domain incremental prompt pool for the continual person searching.

**Strengths:**

The paper provides the first attempt at continual person search.

The performance and ablation studies are effective and detailed.

**Limitations:**

The paper mainly applies existing methods for the continual person search task.

One of the major contributions is to concatenate an existing transformer backbone with a simple feature pyramid for person detection pre-training, which seems limited for publication.

The figures are redundant with exceeded space. Furthermore, the paper could benefit from a more detailed explanation in the caption and a clearer network structure.

**Suitability:**

2

---

### Official Review · Reviewer_zg1p · 2024-05-27

**Rating:** 4
**Confidence:** 3

**Summary:**

The paper "Prompting Continual Person Search" introduces a approach to person search that addresses the challenge of continually learning from sequentially arriving data across multiple domains. The authors propose a Prompt-based Continual Person Search (PoPS) model that leverages a compositional person search transformer and a domain incremental prompt pool with diverse attribute matching. The model aims to strike a balance between stability and plasticity, preventing catastrophic forgetting while adapting to new domains. Extensive experiments validate the proposed method, and the authors plan to release the source code upon publication.

**Strengths:**

1. The paper presents a prompt-based continual learning framework designed specifically for person search. The use of compositional transformers and domain-specific prompts to handle diverse datasets is a particularly noteworthy aspect of the approach.

2. By leveraging pre-trained models and training a more lightweight detection sub-network, this approach substantially reduces the computational resources and data requirements needed, addressing a common challenge in large-scale deep learning initiatives.

3. I like the idea of designing  the domain-incremental prompt pool with diverse attribute matching. It provides a solution for addressing domain shifts. The mechanism for selecting prompts based on domain-specific attributes ensures the model can adaptively generalize to new domains without performance degradation.

4.  Extensive experimental results demonstrate the effectiveness and adaptability of the proposed approach across a variety of person search tasks and domains.

**Limitations:**

1. Though the approach is somewhat novel, the implementation complexity of the PoPS model may pose a challenge for practitioners. The reliance on multiple components, such as hierarchical vision transformers, feature pyramids, and prompt pools, could require substantial expertise and computational resources.

2. The approach assumes access to a reasonable number of person detection datasets for pre-training. However, in real-world situations, the availability of such datasets may be limited, and the model's performance in low-data environments remains uncertain.

3. The pledge to release the source code is admirable, but the paper lacks comprehensive guidelines on how to replicate the results. From my perspective, the training process may be unstable and complex. Providing more clear instructions, specifying the datasets used, and detailing the hyperparameter settings would enhance reproducibility.

**Suitability:**

3

---

### Official Review · Reviewer_3KFC · 2024-06-08

**Rating:** 5
**Confidence:** 3

**Summary:**

This work introduces the novel task of continual cross-domain person search, which requires the model to continually learn from data domains that arrive sequentially and to locate and retrieve specific pedestrians from large-scale images across all previously seen domains. Consequently, the model must balance the acquisition of new knowledge with the prevention of forgetting old knowledge during its continual learning process. To achieve this goal, this paper proposes the Prompt-based Continual Person Search (PoPS) framework built upon a compositional person search transformer. This framework first endows the transformer with the ability to localize pedestrians by introducing a Simple Feature Pyramid. Secondly, it enables the model to learn and retain knowledge by domain-specific prompt pools and a diverse attribute matching module.

**Strengths:**

1. This work stands as a pioneering effort, being the first to introduce the concept of continual person search.

2. The diverse matching module proposed in this paper effectively facilitates the matching of cross-domain prompts, enabling person search across multiple domains without requiring knowledge of domain labels. Theoretically, this module can also be applied to Lifelong Person Re-identification tasks, showcasing broader application potential.

3. Through the training of a lightweight Simple Feature Pyramid, they confers the transformer with robust person localization capabilities, obviating the need for computationally expensive retraining procedures—a creative solution indeed.

4. The experimental outcomes presented are both exhaustive and persuasive, serving as testament to the PoPS framework’s exceptional performance in mitigating the phenomenon of catastrophic forgetting.

**Limitations:**

1. The generalization capability of the model warrants further investigation. Although this paper only explores the performance on seen domains, it remains to be seen whether the knowledge accumulated through prompts across domains can further assist the model in achieving good performance on unseen domains.
2. In my view, the Continual Person Search process can essentially be categorized into Person Detection and Person Re-identification. Given the substantial amount of work already conducted in Lifelong Person Re-identification (LReID), the paper does not provide sufficient discourse on this front. Would it be possible to delve into the relationship between this work and relevant LReID studies [1][2][3].
3. Aesthetics and Clarity: The depiction in Figure 2b is marred by bounding boxes partially obscuring the arrows connecting the prompts selection module to each stage, impacting visual clarity. Enriching Figure 2 with supplementary legends—such as icons denoting cosine similarity and direct multiplication—would enhance reader understanding. Additionally, there may be a writing error in Table 1, as two horizontal lines erroneously appear in the last cell of the final row.

Hope author can release their source code for the future research of LReID committee.

[1] Lifelong person reidentification via adaptive knowledge accumulation, CVPR 2021.
[2] Meta reconciliation normalization for lifelong person re-identification, MM 2022.
[3] A Memorizing and Generalizing Framework for Lifelong Person Re-Identification, PAMI 2023.

**Suitability:**

3

---

### Meta-Review · Area_Chair_a9vq · 2024-07-01

**Recommendation:** Accept (Poster)
**Confidence:** 5

**Metareview:**

The paper initially received the following ratings: BA(zg1p), BR(Swey), BR(uZNe), and WA(3KFC). After rebuttal and discussion, the final ratings were: WA(zg1p), BA(Swey), BA(uZNe), and WA(3KFC).

Below are the details of the rebuttal:

After reading the rebuttal, all the Reviewers were satisfied with the response and ultimately gave an acceptance recommendation as their final rating.

The AC carefully reviewed the rebuttal:
1. This work introduces a novel task of continual cross-domain person search, which is a good starting point for exploring the model's lifelong learning ability in the person search community.
2. Although this paper partly leverages existing methods in person search and the paradigm of prompt learning in continual learning to address continual cross-domain person search issues, the overall framework is effective and reasonable.

In sum, after the rebuttal, the AC believes the concerns raised by the reviewers were well-addressed and recommends acceptance of this paper. In the final version, the AC strongly encourages the authors to include all discussions and reference literature about LReID from the rebuttal and improve the presentation to enhance readability for a diverse audience.

---

### Meta-Review · Senior_Area_Chairs · 2024-07-10

**Recommendation:** Accept (Poster)
**Confidence:** 5

**Metareview:**

This paper received mixed ratings initially. After rebuttal, all the reviewers tend to accpept the paper. SAC and AC agree with reviewers and recommend acceptance of the paper.